# Intimate partner violence: A key correlate of women's physical and mental health in informal settlements in Nairobi, Kenya

**Samantha C. Winter** [1]*, **Lena Moraa Obara**[2], **Sarah McMahon**[3]

**1** Columbia School of Social Work, Columbia University, New York, New York, United States of America,
**2** Department of Sociology and Social Work, University of Nairobi, Nairobi, Kenya, **3** Center on Violence
Against Women and Children, Rutgers, The State University of New Jersey, New Brunswick, New Jersey,
United States of America

* scw2154@columbia.edu

Intimate partner violence: A key correlate of
women's physical and mental health in informal
settlements in Nairobi, Kenya. PLoS ONE 15(4):
e0230894. https://doi.org/10.1371/journal.
pone.0230894

Research and Epidemiology BIPS, GERMANY

**Data Availability Statement:** All relevant data are
within the paper and its Supporting Information
file. The data on which this study is based were
originally collected as part of a pilot project

## Abstract

Globally, one billion people live in informal settlements, and that number is expected to triple
by 2050. Studies suggests that health in informal settlements is a serious and growing con-
cern, yet there is a paucity of research focused on health outcomes and the correlates of
health in these settlements. Studies cite individual, environmental and social correlates to
health in informal settlements, but they often lack empirical evidence. In particular, research
suggests that high rates of violence against women (VAW) in informal settlements may be
associated with detrimental effects on women's health, but few studies have investigated
this link. The purpose of this study was to fill this gap by empirically exploring associations
between women's experiences of intimate partner violence (IPV) and their physical and
mental health. Data for this study were collected in August 2018 in Mathare Valley Informal
Settlement in Nairobi, Kenya. A total of 550 randomly-selected women participated in sur-
veys; however, analyses for this study were run on a subpopulation of the women (n = 361).
Multivariate logistic regressions were used to investigate the link between psychological,
sexual, and emotional IPV and women's mental and physical health. Results suggest that
while some socioeconomic, demographic, and environmental variables were significantly
associated with women's mental and physical health outcomes, all types of IPV emerged
key correlates in this context. In particular, women's experiences of IPV were associated
with lower odds of normal-high physical health component scores (based on SF-36); higher
odds of gynecological and reproductive health issues, psychological distress (based on K-
10), depression, suicidality, and substance use. Findings from this study suggest that poli-
cies and interventions focused on prevention and response to VAW in informal settlements
may make critical contributions to improving health for women in these rapidly growing
settlements.

supported by a grant from the Rutgers Global Health Institute and were collected by the author Dr. Winter as part of a postdoctoral research fellowship at Rutgers, The State University of New Jersey.

**Funding:** SCW - This study was supported by a global health seed grant from Rutgers Global Health Institute, Rutgers University. The funder did not play any role in the study design, data collection and analysis, decision to publish, or preparation of the manuscript. There was no additional external funding received for this study.

**Competing interests:** The authors have declared that no competing interests exist.

## Introduction

Around one-quarter of the world's urban population live in informal settlements—and that number is rising [1]. An informal settlement is defined as a residential area that lacks: durable housing, sufficient living and public spaces, access to basic infrastructure and services like water and sanitation, and secure tenancy [1]. These settlements exist in urban contexts around the world in various forms, scales, and locations, and they go by a range of names (e.g., slums, squatter settlements, favelas, barrios bajos) [1]. The formation and expansion of informal settlements are driven by a range of interrelated factors including population growth; rural-urban migration; lack of affordable housing; poor policy, planning, and land management; economic vulnerability; under- and unemployment; discrimination and marginalization; displacement caused by conflict, natural disasters, and climate change; and, in the case of many sub-Saharan African countries, colonialism [1–3].

In sub-Saharan Africa (SSA) more than half of urban dwellers live in informal settlements, and this number will likely triple by 2050 [4]. In Kenya, specifically, approximately 56% of urban dwellers live in informal settlements [2], and the proportion is expected to reach two-thirds by 2030 [5]. Research conducted in Kenya suggests that residents of these settlements have the worst physical and mental health outcomes of any population in the country [6, 7]. Research also suggests women living in these settlements experience higher rates of violence [8], unemployment [9], and poverty [10] than women in other contexts—variables that evidence shows are linked to poor physical and mental health outcomes [11–13]. Some other individual, socioeconomic, political, social and environmental factors have also been identified as potential correlates of health in informal settlements [14–16]. To date, however, there have been few studies empirically investigating women's physical and mental health and their correlates in informal settlements in Kenya. The purpose of this study was to empirically explore correlates of women's physical health and mental health in a large informal settlement in Nairobi—focusing especially on intimate partner violence (IPV) as a potential correlate.

### Women's physical and mental health in informal settlements

Most of the health-related research carried out in informal settlements has focused on child health and mortality, reproductive health, and maternal and prenatal conditions [6, 15–17], with an emphasis on communicable diseases [16]. In one notable exception, Fink, Arku, and Montana [18] found that women living in informal settlements in Ghana report higher levels of health-related quality of life than people living other urban contexts. On the other hand, Gibbs, Govender, and Jewkes [19] found that 72% of women in informal settlements in South Africa reported moderate to high levels of depressive symptomology and 57.9% reported very high levels, compared to only 26.4% of women in a nationally-representative sample [20]. Some research suggests that, due to lack of access to adequate water, sanitation, and hygienic environments in informal settlements, female residents are also especially vulnerable to reproductive tract infections like vaginal and urinary tract infections (UTIs) [21–25]. One study, reporting on the prevalence of UTIs among women in informal settlements in Bangladesh, for example, found that 46% of women in the sample tested positive for a UTI [26].

### Correlates of health in informal settlements

A growing body of literature cites low-quality housing and infrastructure, and a lack of access to water, sewage, garbage collection, health care, and other basic services as factors associated with poor health in these settlements [14, 15]. Individual demographic characteristics such as age, gender, and socioeconomic status have also been linked to poor physical health in informal settlements [14], and stress has been cited as a factor associated with poor mental health

outcomes in these settlements [16]. Still other studies cite everyday hazards combined with rapid urbanization, insecurity, lack of social cohesion, climate change, and inadequate local government response to these hazards as factors associated with health risks in informal settlements [15].

Some research also suggests that lack of educational and employment opportunities and absence of public sector and law enforcement can foster violence and unrest in settlements and expose residents to greater health risks by limiting their ability to easily and safely navigate their environments and manage their health and daily tasks [6]. In particular, a 2008 study carried out in Nairobi, Kenya found that interpersonal violence was the second leading cause of mortality in informal settlements after infectious diseases (i.e., AIDS and tuberculosis) [27]. Violence against women (VAW)—IPV, in particular—can have deleterious effects on women's health and well-being [28, 29]—effects that may be exacerbated in informal settlements [14].

## Violence against women in informal settlements

VAW in lower- and middle-income countries (LMICs) is receiving growing scholarly attention, and several studies have documented prevalence rates of VAW in informal settlements [8, 17, 30–32], and several studies have linked VAW and crime to environmental and living conditions in these settlements [33–35]. Data collected in informal settlements in Nairobi in 2000 reported that 7% of female and male youth had been the victims of violence by a stranger and 16% had been the victim of violence by a partner or family member [36]—rates that have since risen. In a 2011 study carried out by Corburn and Hildebrand [17] in Mathare Valley—a large informal settlement in Nairobi, Kenya—68% of respondents reported violence as a common physical burden faced by women in this settlement. In 2012, Swart et. al. [8] recorded that 85% of women in Kibera, another large informal settlement in Nairobi, reported having experienced some form of violence in their lifetime compared to 39% of women in the general population in Kenya—conjecturing that these higher levels of violence in these settlements are potentially associated with lack of police and social services, limited access to healthcare, overcrowding, and the living conditions of residents. Finally, data collected from informal settlements in eThekwini Municipality, South Africa, in 2016–2017 showed that 65% of women reported physical and/or sexual violence in the past year [37].

When trying to account for the higher levels of VAW in informal settlements, scholars often employ socio-ecological frameworks to highlight the interplay of factors from multiple levels that shape its occurrence [38]. Some research links factors such as poverty in these settlements to high levels of VAW [33, 39, 40]. In addition, lack of durable housing, insecure tenancy, and lack of opportunity for education and employment may exacerbate incidence of 'stress-induced violence' in informal settlements [33]. Women with irregular, low-paid and casual jobs, like those likely to be found in these settlements, are also more likely to experience IPV [33]. Social instability, weak social ties, high levels of mobility, high unemployment for men, and armed conflict are other factors that have been associated with VAW in informal settlements.[33, 40]. In particular, findings suggests women in these settlements tend to have smaller friendship groups and are, consequently more isolated and less likely to respond to or exit from situations of IPV [33]. Some research suggests that psychological factors, such as "impaired masculinity," often learned by observing violence as a normal behavior in childhood, may also be associated with IPV in informal settlements [33]. Still other socio-ecological theories suggest that IPV is a product of cross-cutting gender-power inequalities that are often exacerbated in economically and socially deprived settings such as these settlements [33, 40, 41]. Gender-based inequities at the structural level include fewer opportunities for women for education, employment, and political power [42]. This is accompanied by community and

cultural norms that often reinforce the unequal status of women, with expectations that men play a dominant role within the household and community and VAW is acceptable [42, 43]. Individual and relationship level factors such as childhood experiences of violence, high levels of stress, beliefs and attitudes that condone violence, mental health issues and use of alcohol and drugs can exacerbate the risk of IPV occurring [38].

## Violence against women and health

Numerous studies have explored associations between violence and poor health outcomes for women [28, 29, 44–46]. The majority of articles discussing the link between VAW and health are focused on IPV and mental health with fewer studies focused on physical health [29]. According to recent reviews, mental health outcomes associated with intimate partner violence include depression, posttraumatic stress disorder (PTSD), anxiety, suicide and self-harm, self-perceived mental health and psychological distress, sleep disturbance, and substance use disorders [28, 29]. Physical health outcomes associated with IPV include HIV/AIDS and other sexually transmitted infections; induced abortion; non-fatal injuries; fatal injuries (homicide); self-perceived limitations in physical; chronic physical health conditions such as chronic pain, fatigue, allergies, loss of hearing or eyesight, cardiovascular issues, diabetes, and gastrointestinal conditions; somatoform disorders and psychosomatic complaints; gynecological symptoms; memory loss, problems with concentration, and dizziness [28, 29].

## Violence against women and health in informal settlements

While there is a growing body of literature focused on VAW in informal settlements and a well-established body of literature linking VAW with poor health, there remains a paucity of research focused on the associations between VAW and health in these settlements. This represents a critical gap in the research given that literature suggests women in these settlements experience much higher rates of violence than women in the general population [8]. The purpose of this study was to help fill this gap by exploring potential associations between women's experiences of violence and their mental and physical health in informal settlements in Nairobi, Kenya.

# Methods

The research was guided by the following research question: to what extent, if at all, is intimate partner violence (IPV), associated with women's mental and physical health outcomes in a large informal settlement in Nairobi, Kenya when controlling for other potential correlates and covariates?

## Data and sample

This study used data collected in Mathare Valley Informal Settlement (Mathare) in Nairobi, Kenya in 2018. Although the exact population and boundaries of the settlement are contested, some sources suggest Mathare is home to approximately 200,000 residents living on less than 3-square kilometers of land [47]. The settlement is divided into 11 villages. A sample of 550 women was selected using a stratified, random sampling technique. Geographic information systems (GIS) was used to randomly select 50 households from each of Mathare's 11 villages and, subsequently, Kish methodology was used to randomly select one woman for each household [48]. To be included in this study, women had to be at least 18 years old and have resided in Mathare for a minimum of six months. Because a key variable of analysis was IPV, analyses for this study were run on a sub-population of the sample (n = 361). This subpopulation

included all women who were married, living with a partner; in a relationship, but not living with a partner; recently separated, divorced, or widowed; or in a casual relationship (had a boyfriend) at some point in the 12 months leading up to the survey. All participants provided verbal informed consent. Verbal consent was deemed appropriate by the Internal Review Board at Rutgers University because a written consent document would have been the only identifying piece of information linking otherwise anonymous surveys to participants (US regulation 45 CFR 46.117). If participants consented, the survey interviewer signed a document affirming consent was given.

Surveys were conducted by female residents in Mathare who were trained on the principles of ethical research and rigorous data collection. Given the sensitive nature of the topic, women were also trained according to the World Health Organization's (WHO) ethical and safety recommendations for research on VAW and sensitive topics [49, 50]. In accordance with these recommendations, investigators, field staff, and a local collaborative board agreed on safety protocols in the event that a participant reported violence and/or adverse mental health outcomes including depression and suicidality [51]. Participants who reported violence or adverse mental health were covertly provided referral to relevant services. In addition, a local counselor was contracted to be in the field throughout data collection. The study was approved by ethics committees at Rutgers, The State University of New Jersey, and the National Commission on Science, Technology, and Innovation in Nairobi, Kenya.

## Measures

**Mental health.**    Women's perceived mental health was measured using a Swahili version of the 36-Item Short Form Health Survey (SF-36). The four domains that make up the mental health factor of the SF-36, i.e., vitality, social functioning, limited emotional functioning, and emotional well-being/mental health subscales were aggregated into a single mental health component score [52]. In a method used by other researchers [53], the mental health component scores were then dichotomized using the median as a cut-off. Participants with a mental health score below the median cut-off were defined as having a low mental health component score while patients with a summary score above or equal to the median cut-off were defined as having a normal-high mental health component score. Women's psychological distress was measured using a Swahili version of the Kessler Scale of Psychological Distress (K10) [54], and those scores were also dichotomized using the median-cut-off method we used for the SF-36 mental health component scores. A Swahili version of the Patient Health Questionnaire-9 Depression Scale (PHQ-9) was used to measure major depressive disorder (MDD) [55]. Respondents were also asked questions about whether they had, *in the last 2 weeks*, experienced suicide ideation or had *ever* attempted suicide. Reponses to these variables were combined into a single 'suicidality' variable for which participants were given a score of '1 = has had suicidal thoughts in the past 2 weeks and/or has ever attempted suicide' or '0 = had never attempted suicide nor had suicidal ideation in the past 2 weeks.' Finally, participants were asked if they *currently* use alcohol and/or tobacco.

**Physical health.**    Women's perceived physical health was measured using the remaining four subscales of the Swahili version of the SF-36 including the physical functioning, role physical functioning, pain, and general health subscales [52]. Again, scores on these sub-scales were aggregated into a single physical health component score and then dichotomized using the median-cut-off method. Additionally, reproductive health and gynecological outcomes were measured using a series of binary yes/no questions about whether women had received medical diagnoses of common conditions such as urinary tract infections, vaginal infections, hemorrhoids, candidiasis/yeast infection, e-coli infection, vaginitis, and/or bacterial vaginosis in

the previous 12 months. The frequencies for some of these conditions were not large enough to analyze separately; thus, a single "reproductive health issues" variable was created for which a woman was given a score of '1 = received medical diagnosis of at least one of these conditions in the last 12 months' and '0 = did not receive medical diagnosis of any of these conditions in the last 12 months.'

**Violence against women.**   This study used a modified version of the domestic violence module used in the Demographic and Health Surveys (DHS) [56]. The measure explores three types of violence (psychological, physical and sexual) perpetrated by an intimate partner. IPV scores were based on women's report of their experience of violence within the past 12 months. The emotional IPV sub-scale consisted of three items, the physical IPV scale had seven items, and the sexual IPV scale consisted of three items. Responses to each of the IPV items were dichotomized. A score of '1' on a sub-scale of violence (psychological, physical and sexual) reflected at least one affirmative response to questions on that sub-scale. A score of '0' reflected the absence of any affirmative response to questions on the sub-scale.

**Sociodemographic and demographic characteristics.**   Finally, a number of sociodemographic variables including household income (measured as a respondent's estimate of monthly income in KES), aware of household finances, education, marital status, number of children, age, employment and having a business were included in the models.

**Environmental factors.**   Research has shown that lack of access to water and sanitation may be associated with poor health outcomes for women in informal settlements [17, 23, 25, 57, 58]. Access to water and sanitation are also used as alternative or additional sociodemographic or wealth measures in these settlements [59, 60]. Thus, women were asked about their primary source for drinking water and sanitation in 24-hr period, and these variables were included in the models.

## Analysis strategy

Descriptive statistics provided information about the study sample and the prevalence rates of IPV in Mathare. In order to answer the key research questions in this study, multivariate logistic regressions were run in Stata statistical software (version 15) [61] to look at associations between women's experiences of recent (in the last 12 months) IPV and their mental and physical health, controlling for other potential sociodemographic and environmental correlates. Two multivariate logistic regression models (Models 1 and 2) were run for *each* of the key physical health outcomes (1. dichotomized physical health component score from the SF-36 and 2. reproductive health issues) and *each* of the mental health outcomes (1. dichotomized mental health component score from the SF-36, 2. dichotomized psychosocial distress from the K10, 3. MDD, 4. suicidality, 5. alcohol and 6. tobacco use). In Model 1, only sociodemographic and environmental variables were included in the multivariate logistic regression. In Model 2, the IPV variables were added to the multivariate logistic regressions from Model 1. Wald tests were used to compare model fit between Models 1 (sociodemographic and environmental variables only) and 2 (IPV variables added to Model 1). Results from the Wald tests are presented in S1 Appendix.

In order to account for the clustered nature of the sample, the descriptive statistics and models were adjusted using the complex survey commands (svy) in Stata. Specifically, villages were used as a clustering variable in the svyset function in Stata. Additionally, because the analysis was conducted on a sub-population of the data (women in a long-term or casual relationship at some point in the past 12 months, n = 360), all analyses, including the descriptive statistics and logistic regressions were adjusted according to the svy and subpop() functions in Stata.

## Results

### Sample characteristics

A summary of the characteristics of the sample is provided in Table 1. According to these results, about 80% of the women in the sample were between the ages of 25 and 44 years and more than half of the women had at least completed primary education. Just over half the women were employed and almost three-quarters of the sample were married. Only 14% of the sample did not have children and close to half of the women had household incomes less than KES 10,000 per month (about $100). Average scores on the SF-36 measures were relatively high compared to women in informal settlements in Ghana [18], but were lower than scores for women in similar settings in Tanzania [52]. Almost 46% of women reported having been diagnosed with at least one reproductive-related health condition in the last 12 months. Almost 18% of women met the criteria for MDD, over 24% had experienced suicidal thoughts in the previous 2 weeks and/or had attempted suicide at least once in their lifetime. Just under two-thirds of women (66.2%) had experienced some form of IPV in the past year.

### Correlates of women's health

Results from the multivariate logistic regression of women's mental health outcomes on potential socioeconomic, environmental and IPV measures are summarized in Table 2. More detailed results including confidence intervals (95%) for both Models 1 and 2 are shown in Table A in S1 Appendix. Similarly, results from the multivariate logistic regressions of women's physical health outcomes on potential correlates are summarized in Table 3 and detailed in Table B in S1 Appendix. Findings suggest that while some sociodemographic and environmental variables were significantly associated with mental and physical health outcomes, the IPV variables emerged as key correlates of women's physical and mental health in Mathare.

**Associations between violence and women's mental health.** IPV was not associated with respondent's mental health component score from the SF-36. Recent psychological IPV was associated with about two and one-half times the odds of meeting the criteria for MDD (OR = 2.6, p<0.01), suicidality (OR = 2.4, p<0.05), and alcohol use (OR = 2.6, p<0.05) and close to four times the odds of tobacco use (OR = 3.8, p<0.05). Sexual IPV was associated with almost two and one half times the odds of having normal-high psychosocial distress (OR = 2.4, p < .001). Finally, physical IPV was associated with about three times the odds of meeting the criteria for MDD and almost four times the odds of suicidality (OR = 3.7, p<0.001). Furthermore, results from the Wald tests comparing Models 1 and 2 (shown in Table B in S1 Appendix) suggest that including IPV variables in the dichotomized psychosocial distress (K10, MDD, suicidality, and alcohol use models significantly improves model fit, but does not improve model fit for the dichotomized mental health (SF-36) or tobacco use models.

**Associations between violence and women's physical health.** Physical IPV was associated with 64% lower odds of having a normal-high physical health (SF-36). Sexual IPV was associated with just under two-times the odds of having been diagnosed with a recent reproductive health condition (OR = 1.97, p<0.05). Results from the Wald tests comparing Models 1 and 2 (shown in Table A in S1 Appendix) further suggest that including IPV variables in the dichotomized physical health (SF-36) and reproductive health conditions models significantly improves model fit.

## Discussion

This study adds to the literature demonstrating poor mental and physical health outcomes for women in informal settlements in Nairobi. Specifically, the results suggest that while some sociodemographic and environmental variables were significantly associated with women's

**Table 1. Adjusted characteristics of sample (n = 361).**

| Socioeconomic demographic variables | Adjusted proportions |
|---|---|
| Age | |
| 18–24 | 13.3 |
| 25–34 | 44.6 |
| 35–44 | 35.5 |
| 45–54 | 6.6 |
| Education | |
| Less than complete primary school | 42.9 |
| Completed primary school, no secondary | 21.6 |
| At least some secondary school | 35.5 |
| Employed | 49.9 |
| Respondent has a business | 34.4 |
| Marital status | |
| Married | 69.0 |
| Living with a man, not married | 7.8 |
| Regular partner, live apart | 18.0 |
| Casual boyfriend | 5.3 |
| Number of children | |
| None | 13.6 |
| 1–2 children | 46.3 |
| 3–4 children | 29.9 |
| 5 or more children | 10.3 |
| Monthly household income | |
| Less than 10,000 KES/month | 47.9 |
| 10,000–15,000 KES/month | 31.0 |
| More than 15,000 KES/month | 21.1 |
| Aware of household income | 91.1 |
| Has access to a toilet at all times | 40.7 |
| Primary drinking water source | |
| Inside tap/well | 7.2 |
| Outside tap/well | 36.0 |
| Public tap/well | 47.9 |
| Vendor/tanker/burst pipe | 8.9 |
| **Physical health outcomes** | |
| Normal-high physical health (from SF-36) | 54.0 |
| Reproductive Issues | 45.7 |
| **Mental health** | |
| Normal-high mental health (from SF-36) | 50.4 |
| Normal-high psychosocial distress (K-10) | 50.1 |
| MDD | 17.7 |
| Suicidality | 24.4 |
| Alcohol use | 21.1 |
| Tobacco use | 7.8 |
| **Intimate partner violence** | |
| Psychological violence | 52.6 |
| Sexual injury from IPV | 42.1 |
| Physical injury from IPV | 64.5 |

**Table 2. Summary of adjusted odds ratios from logistic regression of women's mental health on their experiences of IPV and other potential correlates and covariates.**

| | Binary Mental Health Score | Major Depressive Disorder (MDD) | Suicidality | Alcohol Use | Tobacco Use | Binary Psychosocial Distress |
|---|---|---|---|---|---|---|
| Socio-economic variables | | | | | | |
| Monthly household income | | | | | | |
| 10,000–15,000 KES/month | 0.70 | 1.22 | 1.07 | 2.21* | 8.80** | 0.71 |
| More than 15,000 KES/month | 0.75 | 0.35† | 1.02 | 3.86** | 3.75† | 0.55† |
| Aware of household finances | 0.60 | 0.97 | 1.13 | 5.60† | 5.04 | 1.74 |
| Education | | | | | | |
| Completed primary school | 1.08 | 1.19 | 1.84 | 0.83 | 1.57 | 1.16 |
| At least some secondary | 0.90 | 0.54 | 1.02 | 1.47 | 1.15 | 1.03 |
| Marital status | | | | | | |
| Living with a man, not married | 0.74 | 0.48 | 0.59 | 0.76 | 3.58 | 1.21 |
| Regular partner, live apart | 0.78 | 1.30 | 1.56 | 2.25* | 1.96 | 1.03 |
| Casual boyfriend | 0.23* | 2.14 | 1.84 | 3.21* | 3.08 | 1.30 |
| Number of children | | | | | | |
| 1–2 children | 0.82 | 0.63 | 0.51 | 2.26 | 1.07 | 1.14 |
| 3–4 children | 0.37* | 0.75 | 1.01 | 2.17 | 1.36 | 1.12 |
| 5 or more children | 0.31* | 0.81 | 0.81 | 1.88 | 3.96 | 1.23 |
| Age | 1.01 | 1.01 | 1.02 | 1.01 | 0.94 | 1.00 |
| Respondent is employed | 1.24 | 1.18 | 0.88 | 1.12 | 2.04† | 1.02 |
| Respondent has a business | 0.68 | 2.84** | 1.10 | 0.67 | 3.82* | 1.44 |
| Access to toilet | | | | | | |
| Has access to a toilet at all times | 1.77* | 1.21 | 0.97 | 0.52† | 1.05 | 0.52* |
| Access to water | | | | | | |
| outside tap/well | 3.17* | 1.00 | 0.49 | 0.3† | 0.59 | 0.56 |
| public tap/well | 3.46* | 0.34 | 0.31† | 0.69 | 0.50 | 0.41† |
| vendor/tanker/burst pipe | 2.87 | 2.62 | 1.67 | 2.32 | 3.03 | 0.25* |
| Violence Variables | | | | | | |
| Intimate partner psychological violence | 0.81 | 2.63* | 2.36* | 2.63** | 3.77* | 1.53 |
| Intimate partner sexual violence | 0.82 | 1.51 | 1.27 | 1.07 | 1.33 | 2.39** |
| Intimate partner physical violence | 0.68 | 3.14** | 3.74*** | 0.82 | 0.80 | 1.05 |

† $p < .1$

* $p < .05$

** $p < .01$

*** $p < .001$

mental and physical health outcomes, psychological, sexual, and physical IPV emerged as key correlates of women's health in this environment. In particular, women's experiences of IPV were associated with lower odds of having normal-high physical health; higher odds of gynecological and reproductive health issues; and higher odds of MDD, suicidality, normal-high psychosocial distress, and substance use. These findings potentially corroborate research that suggests women living in these settlements may have worse health outcomes compared to other populations in Kenya [6]. These results also suggest that the prevalence of IPV in these communities (66.2%) is higher than in the general population (39%) [8] and is likely linked to poor physical and mental health for these women—findings that have implications for health and community-based interventions and policy in these settlements.

**Table 3. Summary of adjusted odds ratios from logistic regression of women's physical health on their experiences of IPV and other potential correlates and covariates.**

|  | Binary Physical Health Score | Reproductive Health |
|---|---|---|
| Socio-economic variables |  |  |
| Monthly household income |  |  |
| 10,000–15,000 KES/month | 0.75 | 0.82 |
| More than 15,000 KES/month | 1.32 | 0.74 |
| Aware of household finances | 2.44† | 1.38 |
| Education |  |  |
| Completed primary school | 1.24 | 1.35 |
| At least some secondary | 0.92 | 0.67 |
| Marital status |  |  |
| Living with a man, not married | 0.63 | 0.78 |
| Regular partner, live apart | 0.78 | 1.28 |
| Casual boyfriend | 0.5 | 1.01 |
| Number of children |  |  |
| 1–2 children | 0.79 | 1.08 |
| 3–4 children | 0.5 | 0.62 |
| 5 or more children | 0.65 | 0.65 |
| Age | 0.98 | 0.99 |
| Respondent is employed | 0.99 | 0.95 |
| Respondent has a business | 0.57* | 1.18 |
| Access to toilet |  |  |
| Has access to a toilet at all times | 1.38 | 0.87 |
| Access to water |  |  |
| outside tap/well | 1.93 | 1.3 |
| public tap/well | 2.26 | 1.69 |
| vendor/tanker/burst pipe | 2.07 | 1.79 |
| Violence Variables |  |  |
| Intimate partner psychological violence | 1.7† | 1.23 |
| Intimate partner sexual violence | 0.83 | 1.97* |
| Intimate partner physical violence | 0.36*** | 0.95 |

† $p < .1$

\* $p < .05$

\*\* $p < .01$

\*\*\* $p < .001$

Results from this study suggest that physical IPV is associated with low physical health scores and that sexual IPV is associated with gynecological and reproductive health issues for women in informal settlements in Nairobi. Both of these findings are consistent with research that suggests that physical IPV is associated with poor self-perceived physical health [see 28, 29] and that sexual IPV is associated with endogenous reproductive tract infections (RTIs) [62] and gynecological problems [63–65]. These findings, while perhaps not surprising, suggest there is a critical need for both IPV prevention and response interventions in these settlements.

Although there is a critical need for IPV prevention in these settlements, the overwhelming majority of evaluation research focused on the prevention of IPV has been carried out in the Global North [66, 67]. Physical and legal challenges, e.g. environmental hazards, land tenure

disputes, and population densities, present different barriers to violence prevention in informal settlements [8, 68]. A small, but growing body of research focused on designing and testing violence prevention interventions in informal settlements in LMICs, mostly in South Africa [38, 40, 42, 69, 70], suggests that structural and behavioral interventions such as gender trainings and small-scale interventions, e.g. micro-loan programs, support groups, and job- and skills-trainings, might be effective strategies to help to empower women and serve as models for larger future projects in these settlements [8]. A review of IPV interventions in LMICs also found that 13 out of 16 studies that evaluated structural interventions found positive effects including decreased IPV [42]. A central facet of structural interventions is addressing the key areas of economic empowerment and gender norms, and a small yet increasing body of evidence points to the effectiveness of these approaches [40]. Although there is a need for more research focused on the specific causes of IPV in Mathare—factors that were beyond the scope of this study—small-scale, structural, and behavioral interventions, such as those tested in informal settlements in South Africa, may provide guidance about interventions that, if adapted to the local setting, might also be effective at helping to prevent violence, particularly IPV, in Mathare and similar settlements in Nairobi.

In addition to prevention, the high prevalence rates of violence in combination with the associated poor physical and mental health outcomes found in this study suggest there is also an urgent need for effective response strategies to help survivors of IPV. Specifically, there is a need to explore interventions to screen for, treat, and manage the physical and mental health outcomes associated with experiencing IPV and to provide safety planning and other advocacy strategies to prevent further occurrence of IPV for victims. According to some studies, healthcare utilization is higher among women who are currently experiencing or have recently experienced IPV [71, 72] while other research suggests that IPV is associated with foregone healthcare (being in need of healthcare services, but not seeking help) [73, 74]. This is likely the case for women experiencing IPV in informal settlements. While some of the women seek healthcare services, others may sustain injuries that affect their overall health status; yet, for a variety of reasons, including shame, embarrassment, fear of disclosure to others, or fear of partner retaliation, lack of access to adequate healthcare facilities, and lack of finances, do not seek healthcare and/or omit the potential cause of their physical and mental health symptoms if they do [73]. For this reason, there is likely a need for a multisectoral response to IPV in these settlements.

Recent research has highlighted the critical role of health systems in responding to and treating health outcomes associated with IPV [75]. The Kenyan Ministry of Health has already taken steps towards recognizing the crucial role of health systems in responding to IPV by developing national guidelines on the management of sexual violence in Kenya [76]; however, the guidelines do not yet address IPV nor recognize the unique healthcare requirements of specific populations, e.g., women in informal settlements. Given the findings in this study that suggest there are potentially serious physical and mental health consequences associated with IPV, there is a need to further involve the healthcare sector in developing appropriate interventions, practices, guidelines, and policies to screen for IPV and provide appropriate advocacy, treatment, and referrals for women seeking healthcare in response to IPV in informal settlements. Additionally, there is a need for further research to explore alternative, low-cost interventions to help expand healthcare services to women experiencing IPV in informal settlements who are unable or unwilling to seek healthcare in hospitals and clinics, this is particularly true for women who experience mental health challenges associated with IPV.

Results from this study suggest that all forms of IPV are associated with negative mental health outcomes and/or substance use. Psychological and physical IPV, in particular, were associated with several poor mental health outcomes including MDD, suicidality, and

substance use. Findings also suggest that sexual IPV was associated with normal-high psychosocial distress. These findings are consistent with research from other contexts that suggests women's experiences of violence, especially IPV, are associated with poor mental health and psychological distress [see 28, 29, 77]. Unfortunately, mental health services are often quite difficult to access in informal settlements. There are approximately 100 psychiatrists serving a population of 45 million in Kenya, and the majority of these, while located in urban areas, work in private practice [78]—charging fees well above those affordable to the majority of households in informal settlements. Consequently, despite high levels of IPV in informal settlements, services for survivors are minimal [79]. Recent research carried out in informal settlements in Nairobi, however, suggest there may be low-cost, effective behavioral interventions that can be carried out by lay community members to treat common mental disorders associated with IPV in these settlements [80, 81]. These findings suggest there may be feasible treatment options for women experiencing mental health disorders associated with IPV in informal settlements, but there is clearly a need for more research. There is, especially, a need to develop and test additional low-cost interventions which can be implemented by lay community members (e.g., community health volunteers, traditional healers, or non-medical pharmacy attendants) and/or to adapt and scale up existent interventions for variety of informal settlement contexts.

While the findings from this study provide an important contribution to the literature on IPV and women's health in informal settlements in Kenya, the study has several limitations. The RTI and gynecological variables used in this study, for example, were measured by asking women whether or not they had received medical diagnoses for any of the conditions in the 12 months leading up to the study. There are several barriers that could prevent women from receiving a medical diagnosis for RTI or gynecological conditions in informal settlements. First, findings from a study carried out in informal settlements in Bangladesh [26] suggest that women may not be able to recognize the symptoms of common gynecological conditions and, therefore, not seek treatment. Second, women in informal settlements may not, due to a number of factors including lack of financial resources, seek medical advice even when they are aware of the symptoms. Finally, even if women recognize the symptoms and seek treatment, there are often limited healthcare services available in these settlements and those that are available may not have adequate diagnostic capacity [82]. Furthermore, in lieu of adequate healthcare services or the ability and willingness to seek treatment at a healthcare facility, many women in informal settlements seek treatment from local chemists or non-traditional healers who may not have the knowledge, capacity, or resources to make a medical diagnosis [82, 83]. Consequently, the rates of reported reproductive health conditions in this study are likely underestimating the actual prevalence rates of these conditions in informal settlements. These findings suggest there may be a need for interventions that help women to better recognize the symptoms of RTIs and gynecological conditions and for more accessible and adequate healthcare services as well as alternative prevention and treatment options for women living in these settlements. In addition, there may be a need for follow-up research that uses objective measures, e.g., urinalysis and urine and vaginal cultures, to capture more accurate prevalence rates of reproductive health conditions in these settlements. These recommendations also extend to other physical and mental health conditions beyond RTIs and gynecological conditions. Interventions to help women recognize or screen for symptoms of mental disorders, for example, may help women to seek treatment faster and improve their overall health and wellbeing. Some research has shown that mobile health (mhealth) interventions may help to expand healthcare services, e.g., screening and diagnostic tools, to areas with limited or poor access to health systems such as informal settlements [84], especially for women experiencing IPV [85].

Additional limitations of the study include, first, that the data are cross-sectional; thus, we cannot make any causal claims about the associations between women's experiences of IPV and their health outcomes in informal settlements in Nairobi. Second, that this study focused on violence that occurred within the past 12 months; yet, we know that previous violence can be associated with poorer, long-term health outcomes as well. This limitation highlights the need for more longitudinal studies on VAW and health, especially in communities where poor health and rates of violence are high, e.g. informal settlements [28]. Third, that while this study used adapted versions of the violence measures from the domestic violence module of the DHS, a common measure for violence, it did not include measures for partner controlling behaviors, which might also affect women's mental and physical health in this context—presenting a potentially important area for future research. Fourth, that while some of the health and violence measures used in this study have been validated in a Swahili context and in Kenya, specifically, others are new, modified, or have not. Fifth, for the purposes of these analyses the physical and mental health components of the SF-36 and the psychological distress scores (K10) were dichotomized. While some information may be lost in the dichotomization of continuous variables, the categorization of these variables allowed for a meaningful and accurate interpretation of the results. Finally, that while this study incorporated a variety of measures to explore different forms of IPV, these measures may not provide a comprehensive illustration of VAW in informal settlements in Kenya. For example, we did not include measures to explore financial abuse, which could play an important role in informal settlements where external factors such as poverty may exacerbate all forms, especially financial, abuse in intimate relationships [33, 39, 40].

## Conclusion

The purpose of this study was to explore whether women's experiences of recent IPV were associated with the physical and mental health of women in informal settlements in Nairobi, Kenya. The deprivations common to urban informal settlements including lack of adequate access to clean water and sanitation, housing, healthcare and emergency services, and transport, pose serious health risks to those who live there. A range of socio-ecological factors have also been associated with higher risk of women experiencing violence, particularly IPV, in these communities, which can have additional deleterious effects on women's mental and physical health. Findings from this study suggest that women not only experience high rates of IPV, but that all forms of IPV are associated with poor physical and mental health outcomes for women in a large informal settlement in Nairobi. These findings suggest that policies and interventions focused on preventing VAW in informal settlements may make critical contributions to improving health for women in these rapidly growing environments. In addition, results from this study highlight a critical need for research focused on adapting existing and designing new interventions focused on low-cost, appropriate, and effective screening, advocacy, and safety planning for women experiencing IPV and treatment for related physical and mental health conditions in and outside of formal healthcare systems in informal settlements.

## Supporting information

**S1 Appendix.**
(DOCX)

**S1 Data.**
(XLS)

## Acknowledgments

We would like to thank Everline Achieng, Christine Adhiambo, Anna Mueni, Shainanzi Kaniza, Julia Njoki Nyambura, Mwanaisha Adhiambo Joel, Nancy Kimeu Wanjiru, Milcah Wambui Gakuru, and Ann Lilian Akinyi for their guidance throughout the research project and their commitment to carrying out ethical data collection. You are a wonderful team. The data on which this study is based were originally collected as part of a pilot project supported by a grant from the Rutgers Global Health Institute and were collected by the author Dr. Winter as part of a postdoctoral research fellowship at Rutgers, The State University of New Jersey.

## Author Contributions

**Conceptualization:** Samantha C. Winter.

**Data curation:** Samantha C. Winter, Lena Moraa Obara.

**Formal analysis:** Samantha C. Winter.

**Funding acquisition:** Samantha C. Winter.

**Investigation:** Samantha C. Winter.

**Methodology:** Samantha C. Winter.

**Project administration:** Samantha C. Winter, Lena Moraa Obara.

**Supervision:** Samantha C. Winter.

**Validation:** Sarah McMahon.

**Writing – original draft:** Samantha C. Winter, Lena Moraa Obara, Sarah McMahon.

**Writing – review & editing:** Samantha C. Winter, Lena Moraa Obara, Sarah McMahon.

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
