## [Decision Letter · Decision Letter 0]

31 Oct 2019

PONE-D-19-26708

Intimate partner and non-partner violence: Key correlates of women’s physical and mental health in informal settlements in Nairobi, Kenya

PLOS ONE

Dear Dr. Winter,

Thank you for submitting your manuscript to PLOS ONE. After careful consideration, we feel that it has merit but does not fully meet PLOS ONE’s publication criteria as it currently stands. Therefore, we invite you to submit a revised version of the manuscript that addresses the points raised during the review process.

Importantly, please provide more theoretical backing including appropriate literature, as well as more detail on the statistical analysis that you conducted, in order to fulfill the respective PLOS One publication criterium. Further details are given in the individual reviews.

We would appreciate receiving your revised manuscript by Dec 15 2019 11:59PM. To enhance the reproducibility of your results, we recommend that if applicable you deposit your laboratory protocols in protocols.io, where a protocol can be assigned its own identifier (DOI) such that it can be cited independently in the future. For instructions see: http://journals.plos.org/plosone/s/submission-guidelines#loc-laboratory-protocols

We look forward to receiving your revised manuscript.

Kind regards,

Hajo Zeeb

Academic Editor

PLOS ONE

Journal Requirements:

a) Did participants provide their written or verbal informed consent to participate in this study?

 "SCW - This study was partially supported by a global health seed grant from Rutgers Global Health Institute, Rutgers University. The funder did not play any role in the study design, data collection and analysis, decision to publish, or preparation of the manuscript.".

Reviewers' comments:

Reviewer's Responses to Questions

**Comments to the Author**

1. Is the manuscript technically sound, and do the data support the conclusions?

Reviewer #1: Yes

Reviewer #2: Partly

2. Has the statistical analysis been performed appropriately and rigorously? 

Reviewer #1: No

Reviewer #2: No

3. Have the authors made all data underlying the findings in their manuscript fully available?

Reviewer #1: Yes

Reviewer #2: Yes

4. Is the manuscript presented in an intelligible fashion and written in standard English?

Reviewer #1: Yes

Reviewer #2: Yes

5. Review Comments to the Author

Reviewer #1: This manuscript on violence against women in informal settlements of Nairobi brings attention to an important social problem that only promises to worsen in coming years. I applaud the authors for their work on this project and the effort undertaken to complete the data collection. However, this manuscript suffers from several major issues that preclude its publication as-is. Adding a theoretical basis, answering some serious methodological/analysis questions (and reanalyzing if needed), and a more robust Discussion would help warrant publication after additional review.

Introduction:

Major:

1. Considering PLOS one is targeted to a general audience, the authors might consider adding a brief primer on what informal settlements are and the major reasons for their existence Kenya. This would help ground non-experts in the need for this research.

2. The paper seem to lack any sort of theoretical grounding. What are the theoretical underpinnings for why violence is so high in informal settlements and what can be done about it? There are many potential theories others have used to situate similar studies- I suggest framing the paper using some sort of theoretical construct appropriate to the setting and weaving it throughout the Discussion as well.

Minor:

1. the second sentence uses semicolons to separate clauses instead of commas

2. fractions should be written as words in formal writing- i.e. “two-thirds”

3. Line 75: how can interpersonal violence be second behind both AIDS and TB? Doesn’t that make it third?

4. Line 82- VAW in the developing world has been receiving serious scholarly attention for decades- I don’t think you can say it’s “beginning”

Methods:

Major:

1. Why include only women currently in a relationship? Women who are separated, divorced, widowed, or in casual relationships within the last 12 months could also have experienced IPV. Restricting the sample to women currently in relationship is not uncommon, but this should be listed in the limitations if the team cannot include women who may have experienced IPV in the past year but are not currently in a relationship

2. The use of “medical diagnosis” for the physical health variable(s) is problematic when studying informal settlements since, by the team’s own admission, medical care is scant in these areas. While other methods of measuring these issues may have been unavailable, it stands to reason that a limitation of the study is an underrepresentation of these diagnoses since many women may be living with them but not have had the chance to have them diagnosed

3. The WHO definition of IPV also includes controlling behaviour as a major form of IPV. Is there a reason the researchers did not include controlling behaviour as a form of violence since it is included in the DHS? .

4. Why did the team choose to include both IPV and NPV? The drivers for these and theoretical underpinnings can be quite different- especially in developing country settings. It would be important to justify the inclusion of both kinds of violence.

5. The analysis strategy needs more detail to ensure the analysis methods are correct. Specifically:

a. How did the authors set up the svyset command- specifically, how did the authors take account of weighting, clustering, and stratification to ensure correct point estimates? If the survey set command does not correctly account for these things and multilevel modeling techniques are not used, the results could be largely inaccurate.

b. When models are run, what are the control variables included in each model? How many models were run total?

Minor:

1. Line 210: “thus” should be capitalized

2. Line 217 the use of “stratified” should be replaced with “clustered” or “hierarchical” to avoid confusion with pre-determined strata in the sampling technique

Results:

Major:

1. Why did the authors choose to show only the significant results in the Tables?

2. Do these results reflect adjusted b-coefficients and odds ratios? If so, this should be reflected in the title of the Table

Discussion

Major:

1. The Discussion is devoid of any real connection to why informal settlements are such hotspots for poor health outcomes and violence. Why the discussion of help-seeking is interesting, it is a small part of the larger picture for why this study was conducted, why health outcomes are so poor in informal settlements, and what can be done to reduce violence in these areas. As it stands, the discussion is largely a restating of the results and does little to add to where the literature and science should go next.

2. Line 387- policies and interventions like what?

3. Line 390- is the healthcare sector really the best place to situate future interventions considering the extensive strains and thin coverage already experienced in informal settlements? Can the authors provide some evidence for this suggestion?

Reviewer #2: This is an important contribution to the literature and with some targeted work can be revised for the journal.

1. The gaps in the literature are often stated broadly, such as “Few studies have empirically examined the correlates of health in informal settlements.” I think this needs to be toned back throughout the Introduction.

2. In line 85, please also cite:

Hatcher, A. M., H. Stockl, R. S. McBride, M. Khumalo and N. Christofides (2019). "Pathways From Food Insecurity to Intimate Partner Violence Perpetration Among Peri-Urban Men in South Africa." Am J Prev Med 56(5): 765-772.

Baiocchi, M., R. Friedberg, E. Rosenman, M. Amuyunzu-Nyamongo, G. Oguda, D. Otieno and C. Sarnquist (2019). "Prevalence and risk factors for sexual assault among class 6 female students in unplanned settlements of Nairobi, Kenya: Baseline analysis from the IMPower & Sources of Strength cluster randomized controlled trial." PLoS one 14(6): e0213359.

3. I’m not certain the section starting on line 99 is necessary. Instead of a whole paragraph, this could perhaps be summarized in 2-3 sentences?

4. In line 147, is it “at least 50 households” or can the authors provide the exact number?

5. With this sampling technique, can the estimates be considered population-based? If so, that would be an important distinction of this study vs the extant literature on VAW in informal settlements. Would highlight that throughout

6. I think the DHS items on VAW are accessible widely and do not need to be listed in full in the text. However, perhaps a table in a Supplementary Appendix would be valuable?

7. What is the timeframe for the VAW questions (ever or past 12-months)? How was household income assessed? And how about access to water and sanitation?

8. How were the linear and logistic models built? Did the linear regression outcomes meet assumptions of normality or did they require transformation?

9. In the ethics section please also mention how researchers were trained to assess mental health and violence exposure, and what steps were taken in the cases of current VAW or major depression / suicidality.

10. This is a personal preference (so feel free to ignore) but perhaps Table 1 socio-demographic variables could be dichotomized to save space?

11. Given that non-partner violence has weaker association with the outcomes of interest, I would consider only examining IPV in this paper, and combining the IPV variable to be “ever physical and/or sexual IPV”.

12. To simplify the analysis and make more of a statement about how violence frames outcomes for women, could you combine all the SF-36 items into a single continuous outcome (“physical health”) and say “Any sexual or reproductive health problems”? That way, Table 3 would only have two columns and be easier to interpret. Similarly, I might be tempted to drop Alcohol and Tobacco in Table 2, combine suicidal ideation and attempts into one column (“suicidality”) and report on the Short Form Health survey as one single continuous outcome.

13. Would start Discussion with your own findings, rather than citing literature. You have already made the case for the need for new research, so don’t need to do it again here. First paragraph should sum up your key findings.

14. In line 343 would cite:

Hatcher, A. M., A. Gibbs, R. Jewkes, R. S. McBride, D. Peacock and N. Christofides (2019). "Effect of childhood poverty and trauma on adult depressive symptoms among young men in peri-urban South African settlements." Journal of adolescent health 64: 79-85.

15. The Conclusion could be a bit shorter, and may not require citations (again, this is stylistic). Be sure there are no new ideas introduced here and it’s rather a summary of what’s already been stated.

6. PLOS authors have the option to publish the peer review history of their article (what does this mean?). If published, this will include your full peer review and any attached files.

Reviewer #1: No

Reviewer #2: Yes: Abigail M Hatcher

---

## [Author Response · Author response to Decision Letter 0]

19 Dec 2019

Response to Reviewers’ Comments

We want to thank both the reviewers for the detailed comments and recommendations. We have made a number of changes to the paper. Please note: line numbers referenced here refer to the clean document (with all track-changes accepted).

Reviewer #1

Introduction

Comment 1: 

Considering PLOS one is targeted to a general audience, the authors might consider adding a brief primer on what informal settlements are and the major reasons for their existence Kenya. This would help ground non-experts in the need for this research.

Response:

We have provided a very brief primer on informal settlements in the introduction section with specific reference to some resources for additional reading. See lines 26-36.

Comment 2: 

The paper seems to lack any sort of theoretical grounding. What are the theoretical underpinnings for why violence is so high in informal settlements and what can be done about it? There are many potential theories others have used to situate similar studies- I suggest framing the paper using some sort of theoretical construct appropriate to the setting and weaving it throughout the Discussion as well.

Response:

The reviewer makes an important point. To date there is still relatively little research focused on violence against women in informal settlements, particularly the theoretical underpinnings of why violence is high in these settlements. That being said, we have provided a discussion of the theory presented in literature thus far to help explain the high prevalence of violence in informal settlements in the introduction (see lines 99-121) and throughout the discussion (see, for example, lines 326-344).

Comment 3: 

The second sentence uses semicolons to separate clauses instead of commas

Response:

We have made the corrections (see lines 27-29).

Comment 4: 

Fractions should be written as words in formal writing- i.e. “two-thirds”

Response:

We have made the corrections throughout.

Comment 5: 

Line 75: how can interpersonal violence be second behind both AIDS and TB? Doesn’t that make it third?

Response:

The measures used in the burden of disease study utilized a single category for AIDS and Tuberculosis. As they explain it, “AIDS and tuberculosis were combined in the analysis because about 35% of deaths from AIDS and tuberculosis were due to a probable combination of HIV/AIDS and TB, and hence were coded as "AIDS with TB," while seven cases were coded as "unspecified TB/AIDS." We tried to rephrase the reference in such a way as to reduce confusion for readers (see line 77-79).

Comment 6: 

Line 82- VAW in the developing world has been receiving serious scholarly attention for decades- I don’t think you can say it’s “beginning”

Response:

We have made a correction to help clarify the statement. Please see line 83 for revision.

Methods

Comment 7: 

Why include only women currently in a relationship? Women who are separated, divorced, widowed, or in casual relationships within the last 12 months could also have experienced IPV. Restricting the sample to women currently in relationship is not uncommon, but this should be listed in the limitations if the team cannot include women who may have experienced IPV in the past year but are not currently in a relationship

Response:

In the study all women who reported having had any type of relationship, long-term or casual (e.g., married, living with a partner; in a long-term relationship; separated, divorced, or widowed within the last 12 months, or in casual relationship), at any point in the 12 months leading up to the survey were asked question related to violence. All of these women (n=361) were included in the sample. We have rewritten the language in the manuscript to more accurately reflect the measure (see lines 162-166)

Comment 8: 

The use of “medical diagnosis” for the physical health variable(s) is problematic when studying informal settlements since, by the team’s own admission, medical care is scant in these areas. While other methods of measuring these issues may have been unavailable, it stands to reason that a limitation of the study is an underrepresentation of these diagnoses since many women may be living with them but not have had the chance to have them diagnosed

Response:

We have added several statements and citations to the discussion of limitations associated with using the self-report, medical diagnosis variables to capture information about women’s experiences of reproductive tract and gynecological conditions. Please refer to lines 394-423. 

Comment 9: 

The WHO definition of IPV also includes controlling behaviour as a major form of IPV. Is there a reason the researchers did not include controlling behaviour as a form of violence since it is included in the DHS?

Response:

Unfortunately, the partner controlling behavior questions from the DHS module were not included in the measures of IPV used in this study. The survey was meant to provide preliminary information on the state of women’s physical and mental health, including their experiences of violence in informal settlements. Not all of the questions could be retained in the final version of the survey. Women from the community were involved in the process of creating and cutting down the survey. We have included a statement in the limitations of this study to acknowledge that the controlling behavior by partner questions from the DHS domestic violence module were not included in this study. See lines 430-434.

Comment 10: 

Why did the team choose to include both IPV and NPV? The drivers for these and theoretical underpinnings can be quite different- especially in developing country settings. It would be important to justify the inclusion of both kinds of violence.

Response:

We appreciate all reviewers’ suggestion to focus on IPV. We have, therefore, narrowed the scope of the study to include only the IPV correlates.

Comment 11: 

The analysis strategy needs more detail to ensure the analysis methods are correct. Specifically:

a. How did the authors set up the svyset command- specifically, how did the authors take account of weighting, clustering, and stratification to ensure correct point estimates? If the survey set command does not correctly account for these things and multilevel modeling techniques are not used, the results could be largely inaccurate.

Response:

We appreciate the recommendation for more detail about the analytic strategy and the use of the svy functions in Stata. We have provided a much more detailed description of our models, analytic strategy, and use of the svy functions in our analysis strategy section (see lines 243-248). We are happy to provide additional information if there are still gaps in our description.

b. When models are run, what are the control variables included in each model? How many models were run total?

Response:

We have provided a more detail description of the exact models that were run in the analytic strategy. We have also included detailed results from all of the models in the Appendix so that readers have more information about the composition of the models, model fit, and findings (see Tables A and B in Appendix A).

Comment 12: 

1. Line 210: “thus” should be capitalized

Response:

The correction has been made (see line 227).

Comment 13: 

Line 217 the use of “stratified” should be replaced with “clustered” or “hierarchical” to avoid confusion with pre-determined strata in the sampling technique

Response:

The correction has been made (see line 243). 

Results

Comment 14: 

Why did the authors choose to show only the significant results in the Tables?

Response:

We chose to present on the significant results in the table in the body of the paper because they were much easier to read; however, we modified the tables to include a summary of both significant and non-significant results. We have also included tables with all key information, including confidence intervals in Appendix A. Hopefully this will ensure that all readers have access to the relevant information. Please see Appendix A.

Comment 15: 

Do these results reflect adjusted b-coefficients and odds ratios? If so, this should be reflected in the title of the Table

Response:

All descriptive statistics are adjusted for the clustered nature of the data and the subpopulation analysis. All coefficients are adjusted for the clustered nature of the data and the subpopulation analysis as well as the covariates in the models. These are now reflected in the titles of Tables 1, 2, and 3 and in the column headings in Tables A and B in the Appendix.

Discussion

Comment 16:

The Discussion is devoid of any real connection to why informal settlements are such hotspots for poor health outcomes and violence. Why the discussion of help-seeking is interesting, it is a small part of the larger picture for why this study was conducted, why health outcomes are so poor in informal settlements, and what can be done to reduce violence in these areas. As it stands, the discussion is largely a restating of the results and does little to add to where the literature and science should go next.

Response:

We appreciate this comment. We have, at the request of both reviewers, rewritten a large portion of the discussion and conclusion sections of the paper. We have made considerable effort to talk more extensively about the socio-ecological factors that might contribute to poor health and higher rates of violence against women in informal settlements in the introduction/background/literature review section of the paper (see lines 66-121). In the discussion, we have focused on providing a more detailed set of recommendations about what sorts of policies and interventions might be feasible and/or effective in informal settlements for preventing and responding to IPV and its related physical and mental health consequences for women. Please see revised discussion (lines 307-442).

Comment 17: 

Line 387- policies and interventions like what?

Response:

We have tried to provide much more detail about potential policy and interventions strategies that might start to help address the high VAW prevalence rates and related mental and physical health consequences for women in informal settlements throughout the discussion. Please see revised discussion (lines 307-442).

Comment 18: 

Line 390- is the healthcare sector really the best place to situate future interventions considering the extensive strains and thin coverage already experienced in informal settlements? Can the authors provide some evidence for this suggestion?

Response:

We have provided a more detailed discussion about the rationale for including the health sector in preventing and responding to IPV in informal settlements (see, for example, lines 347-375 and 413-422). We have also included additional discussion about alternative strategies to help expand treatment and resources to women who cannot access health clinics and hospitals (see, for example, lines 372-395 and 422-425).

Reviewer #2:

Comment 1: 

The gaps in the literature are often stated broadly, such as “Few studies have empirically examined the correlates of health in informal settlements.” I think this needs to be toned back throughout the Introduction.

Response:

We thank the reviewer for picking up on the redundancy. We removed a number of these sentences from the background section of the paper including in the introduction, the women’s physical and mental health in informal settlements, correlates of health in informal settlements, violence against women in informal settlements, and the violence against women and health in informal settlements sections.

Comment 2: 

In line 85, please also cite:

Hatcher, A. M., H. Stockl, R. S. McBride, M. Khumalo and N. Christofides (2019). "Pathways From Food Insecurity to Intimate Partner Violence Perpetration Among Peri-Urban Men in South Africa." Am J Prev Med 56(5): 765-772.

Baiocchi, M., R. Friedberg, E. Rosenman, M. Amuyunzu-Nyamongo, G. Oguda, D. Otieno and C. Sarnquist (2019). "Prevalence and risk factors for sexual assault among class 6 female students in unplanned settlements of Nairobi, Kenya: Baseline analysis from the IMPower & Sources of Strength cluster randomized controlled trial." PLoS one 14(6): e0213359.

Response:

We have incorporated these additional resources into the paper as suggested. We appreciate that the reviewer brought these important articles to our attention.

Comment 3:

I’m not certain the section starting on line 99 is necessary. Instead of a whole paragraph, this could, perhaps, be summarized in 2-3 sentences?

Response:

We have cut the section down to 4 sentences (see lines 123-134). We are happy to make additional adjustments if that will increase the readability of the paper.

Comment 4: 

In line 147, is it “at least 50 households” or can the authors provide the exact number?

Response:

We have made the correction (see lines 157-160).

Comment 5:

With this sampling technique, can the estimates be considered population-based? If so, that would be an important distinction of this study vs the extant literature on VAW in informal settlements. Would highlight that throughout

Response:

While the estimates may be considered population based, there may be additional sampling considerations that would need to be taken into account to truly assume these are population-based. Thus, for the purposes of this paper, we are not claiming that they are population-based.

Comment 6: 

I think the DHS items on VAW are accessible widely and do not need to be listed in full in the text. However, perhaps a table in a Supplementary Appendix would be valuable?

Response:

We have cut down the description of VAW measures considerably and cited the DHS domestic violence module. Please see lines 210-218.

Comment 7: 

What is the timeframe for the VAW questions (ever or past 12-months)? How was household income assessed? And how about access to water and sanitation?

Response:

The timeframe is IPV in the past 12-months. We have noted this in the measures section (lines 212-213). Measures for household income, access to sanitation, and primary drinking water source were described in the measures section (see lines 220-221 and 226-227).

Comment 8: 

How were the linear and logistic models built? Did the linear regression outcomes meet assumptions of normality or did they require transformation?

Response:

We have provided a more detailed description of our analytic strategy to better explain the process of building the multivariate models used to analyze the data in this study (see lines 229-248). Original versions of some of the SF-36 outcomes did require transformations; however, based on the reviewer’s recommendations to combine/collapse some of the physical and mental health outcome variables, we combined four sub-scales of the SF-36 into an aggregate physical health component score variable and the remaining four sub-scales of the SF-36 into a separate aggregate mental health component score variable as is commonly done with the SF-36. Additionally, these component scores, as well as the remaining continuous variable (psychological distress measured with the K-10 scale), were dichotomized using a median-cut-off method described in detail in the methods section (see lines 179-190 and 198-202) and employed by other researchers using similar methodological approaches with these health outcome variables [1].

Comment 9: 

In the ethics section please also mention how researchers were trained to assess mental health and violence exposure, and what steps were taken in the cases of current VAW or major depression / suicidality.

Response:

We have added more detail about the protocols for handling instances where women reported violence and/or adverse mental health outcomes (see lines 167-177).

Comment 10:

This is a personal preference (so feel free to ignore) but perhaps Table 1 socio-demographic variables could be dichotomized to save space?

Response:

We have dichotomized all of the continuous variables in Table 1, but we did leave the factorial structure for the categorical/nominal variables. The size of the table was greatly reduced, but still contains some level of detail. We are happy to make further adjustments if the table is still cumbersome to read. 

Comment 11: 

Given that non-partner violence has weaker association with the outcomes of interest, I would consider only examining IPV in this paper, and combining the IPV variable to be “ever physical and/or sexual IPV”.

Response:

We have removed NPV from the analysis. We are focusing solely on IPV. We did, however, decide to keep the three different levels of the IPV variable because they have differential associates with various mental and physical health outcomes.

Comment 12:

To simplify the analysis and make more of a statement about how violence frames outcomes for women, could you combine all the SF-36 items into a single continuous outcome (“physical health”) and say “Any sexual or reproductive health problems”? That way, Table 3 would only have two columns and be easier to interpret. Similarly, I might be tempted to drop Alcohol and Tobacco in Table 2, combine suicidal ideation and attempts into one column (“suicidality”) and report on the Short Form Health survey as one single continuous outcome.

Response:

We appreciate these suggestions about how to simplify the results for this study. We combined four sub-scales of the SF-36 into an aggregate physical health component score variable and the remaining four sub-scales of the SF-36 into a separate aggregate mental health component score variable as is commonly done with the SF-36. We also combined responses from all of the reproductive and gynecological conditions into a single dichotomous variable and presented only these results in the tables and text. We also created the “suicidality” variable as the reviewer suggested. We did, however, keep the alcohol and tobacco use variables in the results as substance use is a health outcome commonly associated with experiences of IPV [2, 3]. Pleased see revised measures sections (see lines 179-209)

Comment 13:

Would start Discussion with your own findings, rather than citing literature. You have already made the case for the need for new research, so don’t need to do it again here. First paragraph should sum up your key findings.

Response:

We have the adjustment to our discussion. In fact, we have, based on comments from both reviewers, rewritten a large portion of the discussion (see lines 307-442).

Comment 14: 

In line 343 would cite:

Hatcher, A. M., A. Gibbs, R. Jewkes, R. S. McBride, D. Peacock and N. Christofides (2019). "Effect of childhood poverty and trauma on adult depressive symptoms among young men in peri-urban South African settlements." Journal of adolescent health 64: 79-85.

Response:

We appreciate the suggestion. We have made the addition.

Comment 15:

The Conclusion could be a bit shorter, and may not require citations (again, this is stylistic). Be sure there are no new ideas introduced here and it’s rather a summary of what’s already been stated.

Response:

We have shortened the conclusion considerably, removed references to literature, and provided more concrete recommendations for future policy and intervention. The discussion also contains considerably more recommendations for policy and intervention strategies to help prevent IPV in informal settlements to respond to physical and mental health outcomes associated with IPV in these communities. See lines 443-459 for revised conclusion.

 

References

1. Sineke T, Evans D, Schnippel K, van Aswegen H, Berhanu R, Musakwa N, et al. The impact of adverse events on health-related quality of life among patients receiving treatment for drug-resistant tuberculosis in Johannesburg, South Africa. Health and quality of life outcomes. 2019;17(1):94. doi: 10.1186/s12955-019-1155-4.

2. Abramsky T, Watts CH, Garcia-Moreno C, Devries K, Kiss L, Ellsberg M, et al. What factors are associated with recent intimate partner violence? findings from the WHO multi-country study on women's health and domestic violence. BMC public health. 2011;11(1):109.

3. Oram S, Khalifeh H, Howard LM. Violence against women and mental health. The Lancet Psychiatry. 2017;4(2):159-70.

---

## [Decision Letter · Decision Letter 1]

23 Jan 2020

PONE-D-19-26708R1

Intimate partner violence: A key correlate of women’s physical and mental health in informal settlements in Nairobi, Kenya

PLOS ONE

Dear Dr. Winter,

Thank you for submitting your manuscript to PLOS ONE. After careful consideration, we feel that it has merit but does not fully meet PLOS ONE’s publication criteria as it currently stands. Therefore, we invite you to submit a revised version of the manuscript that addresses the remaining few points raised during the review process.

Please ensure that the issue of dichotomizing the SF-36 is adressed in your revised version.

We would appreciate receiving your revised manuscript by Mar 08 2020 11:59PM. To enhance the reproducibility of your results, we recommend that if applicable you deposit your laboratory protocols in protocols.io, where a protocol can be assigned its own identifier (DOI) such that it can be cited independently in the future. For instructions see: http://journals.plos.org/plosone/s/submission-guidelines#loc-laboratory-protocols

We look forward to receiving your revised manuscript.

Kind regards,

Hajo Zeeb

Academic Editor

PLOS ONE

Reviewers' comments:

Reviewer's Responses to Questions

**Comments to the Author**

1. If the authors have adequately addressed your comments raised in a previous round of review and you feel that this manuscript is now acceptable for publication, you may indicate that here to bypass the “Comments to the Author” section, enter your conflict of interest statement in the “Confidential to Editor” section, and submit your "Accept" recommendation.

Reviewer #1: (No Response)

2. Is the manuscript technically sound, and do the data support the conclusions?

Reviewer #1: Yes

3. Has the statistical analysis been performed appropriately and rigorously? 

Reviewer #1: Yes

4. Have the authors made all data underlying the findings in their manuscript fully available?

Reviewer #1: Yes

5. Is the manuscript presented in an intelligible fashion and written in standard English?

Reviewer #1: Yes

6. Review Comments to the Author

Reviewer #1: This manuscript focuses on the correlates of intimate partner violence (IPV) among women in informal settlements of Nairobi. I thank the authors for their conscientious responses to the reviewer comments and feel the paper is much improved. I do have a couple of comments below, but will leave it to the editor to ascertain whether these constitute another round of revisions of if the paper can be provisionally accepted.

The paper is much improved and sets the stage well for the analysis to follow. A few notes:

a. Line 83- “developing countries” (which I admit I used in my own review of this paper) may not be the most appropriate term. Check with PLOS for their preferred term to refer to what are commonly referred to “low- and middle-income countries” or LMIC.

b. Line 138- the same sentence regarding the Gibbs article from earlier in the introduction is mentioned again here- one reference is likely sufficient

c. Line 234: a comma should be placed after “study”

d. I am concerned that substantial information is being lost in the SF-36 by dichotomizing at the median. Is there a reason the authors decided to dichotomize instead of leaving it as a continuous variable and using linear regression for these models? A sentence expounding on this decision prior to its justification using prior studies would be helpful.

7. PLOS authors have the option to publish the peer review history of their article (what does this mean?). If published, this will include your full peer review and any attached files.

Reviewer #1: No

---

## [Author Response · Author response to Decision Letter 1]

5 Feb 2020

Response to Reviewers’ Comments

We want to thank the reviewer, again, for the insightful recommendations. We have made the appropriate adjustments to the paper and responded to the specific comments below. Please note: line numbers referenced here refer to the clean document (with all track-changes accepted).

Reviewer #1

Comment 1:

Line 83- “developing countries” (which I admit I used in my own review of this paper) may not be the most appropriate term. Check with PLOS for their preferred term to refer to what are commonly referred to “low- and middle-income countries” or LMIC.

Response:

Thank you for this comment. We have made the adjustment to use LMIC. (See ln 83)

Comment 2:

Line 138- the same sentence regarding the Gibbs article from earlier in the introduction is mentioned again here- one reference is likely sufficient

Response:

We have made the adjustment.

Comment 3:

Line 234: a comma should be placed after “study”

Response:

We have make the adjustment.

Comment 4:

I am concerned that substantial information is being lost in the SF-36 by dichotomizing at the median. Is there a reason the authors decided to dichotomize instead of leaving it as a continuous variable and using linear regression for these models? A sentence expounding on this decision prior to its justification using prior studies would be helpful.

Response:

We really appreciate this point and spent a great deal of time considering our approach to this issue both in the first revision and in this one. While some information may be lost in the dichotomization of the SF-36 at the median, the combined physical health and mental health components are not normally distributed and cannot be transformed in such a way to ensure the normality assumptions of a linear regression are being met; thus, the findings in the linear regression models may be biased or not interpretable. Following previous methods employing the SF-36, we felt the best approach to ensure the results were meaningful and accurate was to dichotomize the scores [1-3]. However, we have also acknowledged this limitation in the limitations section (please see lines 428-431).

 

References

1. Casso D, Buist DS, Taplin S. Quality of life of 5–10 year breast cancer survivors diagnosed between age 40 and 49. Health and quality of life outcomes. 2004;2(1):25.

2. Cioncoloni D, Innocenti I, Bartalini S, Santarnecchi E, Rossi S, Rossi A, et al. Individual factors enhance poor health-related quality of life outcome in multiple sclerosis patients. Significance of predictive determinants. Journal of the neurological sciences. 2014;345(1-2):213-9.

3. Sineke T, Evans D, Schnippel K, van Aswegen H, Berhanu R, Musakwa N, et al. The impact of adverse events on health-related quality of life among patients receiving treatment for drug-resistant tuberculosis in Johannesburg, South Africa. Health and quality of life outcomes. 2019;17(1):94. doi: 10.1186/s12955-019-1155-4.

---

## [Editor Report · Decision Letter 2]

12 Mar 2020

Intimate partner violence: A key correlate of women’s physical and mental health in informal settlements in Nairobi, Kenya

PONE-D-19-26708R2

Dear Dr. Winter,

We are pleased to inform you that your manuscript has been judged scientifically suitable for publication and will be formally accepted for publication once it complies with all outstanding technical requirements.

With kind regards,

Hajo Zeeb

Academic Editor

PLOS ONE
---

## [Editor Report · Acceptance letter]

18 Mar 2020

PONE-D-19-26708R2 

Intimate partner violence: A key correlate of women’s physical and mental health in informal settlements in Nairobi, Kenya 

Dear Dr. Winter:

I am pleased to inform you that your manuscript has been deemed suitable for publication in PLOS ONE. Congratulations! Your manuscript is now with our production department. 

With kind regards,

on behalf of

Prof. Hajo Zeeb 

Academic Editor

PLOS ONE